# MultiFuseYOLO: Redefining Wine Grape Variety Recognition through Multisource Information Fusion

**DOI:** 10.3390/s24092953

**Published:** 2024-05-06

**Authors:** Jialiang Peng, Cheng Ouyang, Hao Peng, Wenwu Hu, Yi Wang, Ping Jiang

**Affiliations:** 1College of Information and Intelligence, Hunan Agricultural University, Changsha 410128, China; pjl98@stu.hunau.edu.cn (J.P.); ouyang@stu.hunau.edu.cn (C.O.); sx20230201@stu.hunau.edu.cn (H.P.); 2College of Mechanical and Electrical Engineering, Hunan Agricultural University, Changsha 410128, China; teacher_jp@163.com

**Keywords:** wine grape, SynthDiscrim, multisource, WineYOLO-RAFusion, MultiFuseYOLO

## Abstract

Based on the current research on the wine grape variety recognition task, it has been found that traditional deep learning models relying only on a single feature (e.g., fruit or leaf) for classification can face great challenges, especially when there is a high degree of similarity between varieties. In order to effectively distinguish these similar varieties, this study proposes a multisource information fusion method, which is centered on the SynthDiscrim algorithm, aiming to achieve a more comprehensive and accurate wine grape variety recognition. First, this study optimizes and improves the YOLOV7 model and proposes a novel target detection and recognition model called WineYOLO-RAFusion, which significantly improves the fruit localization precision and recognition compared with YOLOV5, YOLOX, and YOLOV7, which are traditional deep learning models. Secondly, building upon the WineYOLO-RAFusion model, this study incorporated the method of multisource information fusion into the model, ultimately forming the MultiFuseYOLO model. Experiments demonstrated that MultiFuseYOLO significantly outperformed other commonly used models in terms of precision, recall, and F1 score, reaching 0.854, 0.815, and 0.833, respectively. Moreover, the method improved the precision of the hard to distinguish Chardonnay and Sauvignon Blanc varieties, which increased the precision from 0.512 to 0.813 for Chardonnay and from 0.533 to 0.775 for Sauvignon Blanc. In conclusion, the MultiFuseYOLO model offers a reliable and comprehensive solution to the task of wine grape variety identification, especially in terms of distinguishing visually similar varieties and realizing high-precision identifications.

## 1. Introduction

Wine grape variety identification poses a significant challenge for the wine industry [1]. Particularly when relying solely on a single grape leaf or fruit for varietal identification, traditional deep learning modeling methods often struggle to effectively distinguish varieties with a high degree of similarity [2]. The complexity of this challenge stems from the diversity of grapes, encompassing variations in morphology, color, size, and varietal characteristics influenced by different growing environments [3]. Traditional deep learning approaches falter in addressing this intricate and variable scenario [4].

To address this issue, a novel deep learning approach is proposed in this study to mitigate the challenges associated with using a single feature for variety identification [5]. By employing deep learning algorithms such as convolutional neural networks (CNNs) and recurrent neural networks (RNNs), a multifeature fusion strategy is introduced, integrating information from both grape leaves and fruits [6]. This methodology deviates from conventional deep learning techniques by emphasizing the amalgamation of information from diverse sources, thereby enhancing the accuracy of identifying similar varieties [7]. The significance of this study is to fill the gap in the existing research on wine varietal recognition and to propose an innovative solution to the limitations of traditional deep learning models in this task. While traditional single-feature-dependent models perform poorly when facing varieties with high similarity, the multisource information fusion approach proposed in this paper provides new ideas to address this challenge. By optimizing and improving the deep learning model and fusing multisource information into the model, this study achieves more accurate identification of similar varieties, thus providing a more reliable varietal identification tool for the wine industry. This research result is of great significance to grape growers, winemakers and wine producers, which can improve the accuracy and efficiency of varietal identification, and thus promote the improvement of wine quality and market competitiveness. Therefore, the work in this paper is not only academically innovative and practical, but also positively contributes to the development of the wine industry.

Deep learning has been extensively applied in varietal recognition across various agricultural and cash crops. Alper Taner [8] utilized a deep learning model for hazelnut varietal classification, achieving an accuracy of 98.63%. Karim Laabassi et al. [9] employed a deep learning approach for wheat variety classification, yielding classification accuracies ranging from 85% to 95.68%. Murat Koklu [10] achieved notable success in the classification of rice varieties using a deep learning approach. Chunguang Bi [11] devised a neural network capable of accurately and efficiently classifying corn seeds, meeting the stringent requirements of high-precision classification of corn seed images. Poornima Singh Thakur [12] demonstrated that their method outperformed several contemporary deep learning methods in crop disease identification, achieving an accuracy of 99.16% on the PlantVillage dataset. Resul Butuner [13] attained the highest classification success rate of 99.80% using an ANN algorithm that leveraged deep features obtained from the SqueezeNet model. Nurhadi Wijaya [14] employed CNN for efficient citrus type classification, achieving a commendable accuracy of 96.0% in classifying oranges. Deep learning methods possess robust feature learning and representation capabilities, enabling the extraction of intricate characteristics of grape varieties from extensive datasets [15]. Bogdan Franczyk [16] developed a vineyard grape recognition model with a 99% accuracy rate in correctly identifying grape varieties that are relatively discernible. Amin Nasiri [17] employed a modified deep learning model that achieved an average classification accuracy of more than 99% in recognizing different grape varieties. Marco Sozzi [18] obtained promising results using YOLOv3, YOLOv4, and YOLOv5 deep learning algorithms for the automatic detection of white grape varieties. Compared to traditional machine learning methods, deep learning exhibits superior performance in handling large-scale data, enabling the extraction of underlying patterns and enhancing model generalization capabilities [19]. In the realm of grape variety identification, deep learning not only enhances identification accuracy but also adapts well to variations in grape varieties across different regions, climates, and growing conditions, thereby exhibiting enhanced robustness [20]. Although deep learning has shown strong potential for grape variety identification, the wine industry still faces many challenges. Varietal mixing and cross-fertilization are more common in the grape growing process, making it more difficult to distinguish between similar varieties [21]. In addition, the morphological characteristics of grape varieties are influenced by the growing environment, climate, and soil conditions, further increasing the complexity of identification [22]. Therefore, there is a need to develop a method to accurately identify grape varieties to meet the needs of the wine industry for quality assurance and product traceability.

Against this background, this study is dedicated to addressing the needs and challenges of varietal identification in the wine industry. Our goal is to construct an efficient and accurate grape variety identification model to provide reliable quality assurance and product traceability services. To address the challenges of recognizing grape varieties, we propose a novel approach that has the following three key contributions.

WineYOLO-RAFusion multiscale fusion network model: This innovative target detection model integrates multiscale fusion techniques to achieve more comprehensive information extraction, thus improving the recognition accuracy of wine grape varieties.

MultiFuseYOLO model: We propose a new model, MultiFuseYOLO, for recognizing wine grape varieties with high similarity, which integrates fruit and leaf features and improves the model’s ability to discriminate between varieties through comprehensive and accurate fusion of information from multiple sources.

SynthDiscrim algorithm: This algorithm is the core algorithm of the multisource information fusion method, which seamlessly integrates fruit and leaf information by using complex techniques such as weighted feature fusion, optimized threshold setting and combined discriminant score calculation.

## 2. Materials and Methods

### 2.1. Dataset

This experiment used the publicly available dataset Embrapa WGISD [23] as the primary experimental dataset Self-harvested leaf dataset as the supplementary experimental dataset. Uses of the Embrapa WGISD include relaxation of the instance segmentation problem, classification, semantic segmentation, object detection and counting, and WGISD can also be used for grape varieties. The dataset consists of 300 images containing 4432 grape bunches labeled with bounding boxes. Of these images, 137 were labeled with a binary mask to identify the pixels of each grape bunch. This means that out of 4432 grape bunches, 2020 grape bunches were provided with binary masks for instance segmentation. The grape berries in WGISD are shown in Figure 1 and the number of images in WGISD is shown in Table 1.

This data collection took place in September 2023 at a wine estate in Yantai, Shandong Province, focusing on grape variety identification. The wine estate specializes in wine grapes, covering five varieties, Noble Aroma, Matheran, Cabernet Sauvignon, Sauvignon Blanc, and Chardonnay. Due to the limitations of the collection season, not all grape plants were fruiting at the time, so we focused our data collection on photographing the leaves.

During data collection, we used a selfie stick with a Sony lens and kept the shooting distance at 20–30 cm. This approach not only facilitated high-quality photography of the target, but also utilized the cell phone for data supplementation, ensuring comprehensive and accurate collection. Camera lens section parameters are shown in Table 2 below. The grape leaves in the self-collected dataset are shown in Figure 2, and the number of images in the self-harvested leaf is shown in Table 3.

As can be seen from the datasets shown above, the publicly available dataset Embrapa WGISD contains mainly grape fruit bunches, so our annotation of it focused mainly on the bunches. The self-collected dataset, on the other hand, mainly contains grape leaves, so our labeling of the self-collected dataset mainly focused on the leaves.

Such dataset characteristics made it necessary for us to focus on the features of different parts of the annotation process, focusing on the morphology and structure of fruit bunches in the case of Embrapa WGISD, while in the case of the self-collected dataset, more attention was paid to the features of leaves and their related information. Such an annotation strategy helped to ensure that our annotation work on different datasets was more precise and targeted.

### 2.2. WineYOLO-RAFusion Multiscale Fusion Network Modeling

First and foremost, we established a deep learning model as the cornerstone of integrated information fusion to distinguish it from other conventional deep learning models [24]. The model had to achieve high localization accuracy while also ensuring robust recognition accuracy for grape varieties.

The WineYOLO-RAFusion model, developed with YOLOv7 [25] as a benchmark, introduces the innovative Res-Attention module [26] and a CFP-centered feature pyramid multiscale fusion module to address the challenges associated with wine grape variety identification and fruit localization tasks.

The model inherits the real-time capabilities and high efficiency of YOLOv7, providing a strong foundation for the target detection task [27]. By incorporating the Res-Attention module, the model introduces residual connections and attention mechanisms during the feature extraction phase, significantly enhancing focus on the target region. This enhancement boosts the model’s sensitivity to grape variety features, thereby further improving the accuracy of variety identification.

Secondly, the WineYOLO-RAFusion model employs a CFP-centered feature pyramid multiscale fusion module to better accommodate the diversity of wine grapes. By constructing a centered feature pyramid, the model can fuse features at different scales in a more focused manner, enhancing its ability to represent grape berries across multiple scales [28]. This renders the model more adaptable to variations in fruit sizes and shapes for the fruit localization task, consequently improving the accuracy and robustness of fruit localization [29].

In summary, the WineYOLO-RAFusion model offers an efficient and accurate solution for varietal identification and fruit localization tasks in wine grapes by integrating YOLOv7, the Res-Attention module, and the CFP-centered feature pyramid multiscale fusion module. This model amalgamates the classical framework of target detection with advanced feature fusion techniques, providing robust support for intelligent production in the winemaking industry.

#### 2.2.1. Res-Attention Module

The Res-Attention module is a new attention mechanism based on the CBAM [30] (convolutional block attention module) module. The Res-Attention module aims to reduce the information loss that may occur during the information transmission process by introducing a residual structure to minimize the information loss that may occur during the transmission process. The structure of the Res-Attention module is shown in Figure 3.

In the Res-Attention module, the feature map first enters the channel attention model, which retains some of the input information [31]. By introducing a spatial attention model after the channel attention model, some of the previously retained information is fused with the information output from the spatial attention module [32]. Eventually, the fused feature information is output to the final output of the Res-Attention module.

The channel attention mechanism first takes the input feature map ${F\in R}^{C\times H\times W}$ and passes it through global maximum pooling and global average pooling to obtain two $\left(C\times 1\times 1\right)$ feature maps ${M_{S}\in R}^{C\times 1\times 1}$. Then, the resulting feature maps pass through a shared network consisting of a two-layer multilayer perceptron (*MLP*) with the number of neurons in the first hidden activation layer C/r, where r is the shrinkage rate, and the number of neurons in the second layer is C. Afterwards, the output features of the shared network are summed element-wise, and the final complete feature vector is output by the channel attention mechanism $M_{C}\left(F\right)$. The formula is shown below.
$$\begin{array}{cc} M_{C}\left(F\right) & =\sigma \left(MLP\left(AvgPool\left(F\right)\right)+MLP\left(MaxPool\left(F\right)\right)\right)\\ & =\sigma \left(W_{1}\left(W_{0}\left(F_{avg}^{C}\right)\right)+W_{1}\left(W_{0}\left(F_{max}^{C}\right)\right)\right) \end{array}$$
where σ denotes the sigmoid function; $W_{0}\in R^{C/r\times C}$, $W_{1}\in R^{C/r\times C}$, and the *MLP* weights $W_{0}$ and $W_{1}$ are shared for both inputs; and $W_{0}$ is after the ReLU activation function.

The spatial attention mechanism first takes the feature vectors obtained from the previous channel attention mechanism module as the input feature vectors for this module. The input feature vector is first subjected to a maximum pooling operation and an average pooling operation to obtain two feature vectors $F_{max}^{S}\in R^{1\times H\times W}$ and $F_{avg}^{S}\in R^{1\times H\times W}$, respectively. Then, the maximum pooled features and average pooled features are subjected to a channel splicing operation. Afterwards, the feature vectors are reduced to one dimension by a convolutional convolution operation of (7 × 7). Finally, after a sigmoid function, the feature vector is obtained, $M_{S}(F)\in R^{H\times W}$. The formula is shown below.
$$\begin{array}{cc} M_{S}\left(F\right) & =\sigma \left(f^{7\times 7}\left(\left[AvgPool\left(F\right);MaxPool\left(F\right)\right]\right)\right)\\ & =f^{7\times 7}\left(\left[F_{avg}^{S};F_{max}^{S}\right]\right) \end{array}$$
where σ denotes the sigmoid function and $f^{7\times 7}$ denotes the convolution operation with a convolution kernel of size (7 × 7).

The design of this new Res-Attention module takes into account information integrity during feature transfer and emphasizes the importance of previously retained information by means of residual concatenation. This design allows the module to capture key features in the image more efficiently and achieve more comprehensive information fusion in both spatial and channel dimensions. The introduction of the Res-Attention module is expected to enhance the performance of the model in image processing tasks, especially in scenarios where a large amount of detailed information needs to be processed and contextual relationships need to be preserved.

#### 2.2.2. FPN Feature Pyramid

The FPN [33] (feature pyramid network) is a deep learning network structure for target detection and semantic segmentation tasks, aiming to efficiently process feature information at different scales to enhance the model’s ability to perceive and localize targets.

The core idea of the FPN is to enable the network to focus on both details and overall information in an image by building a multiscale feature pyramid [34]. Its main components include the bottom feature map, the top feature map, and the lateral connections [35]. The bottom feature map is the original feature map generated by the backbone network (usually a pre-trained convolutional neural network such as ResNet [36] or VGG [37]). The top layer feature map contains the high-level semantic features obtained from the bottom layer feature map through multiple up-sampling and convolutional operations.

Horizontal connectivity is responsible for connecting the bottom feature map to the top feature map in order to form a feature pyramid [38]. In this way, the FPN network is able to capture the multiscale features of the image at different levels, enabling the network to have a good perception of both small and large targets. In the task of target detection, the introduction of the FPN effectively solves the problem of uneven processing of different size targets by the network and improves the accuracy and robustness of target detection.

Overall, the FPN feature pyramid provides deep learning models with more comprehensive visual information by integrating multiple levels of feature information, making them more effective in processing multiscale objects. This design has enabled the FPN to achieve significant performance gains in tasks such as target detection and semantic segmentation, making it an important component in many visual tasks.

#### 2.2.3. CFP Centered Feature Pyramid Multiscale Fusion Module

Based on the structure and functions of the FPN introduced above, the CFP is a module that improves and enhances the FPN. The WineYOLO-RAFusion proposed in this paper, on the other hand, incorporates an FPN-based CFP module. The centralized feature pyramid (CFP) [39] is an innovative target detection algorithm, whose core idea is to introduce a global explicit centralized conditioning scheme to optimize the construction of the feature pyramid and the information extraction process. Compared with the existing methods, CFP not only focuses on the feature interactions between different layers, but also considers the feature adjustment within the same layer, which shows significant advantages especially in the dense prediction task.

This approach proposes a spatially explicit visual center scheme consisting of a lightweight multilayer perceptron (*MLP*) [40] for capturing global remote dependencies, and a learnable visual center for aggregating local critical regions. By globally centralizing the supervision of commonly used feature pyramids in a top-down manner, the CFP effectively enhances the extraction of multiscale feature information and achieves consistent performance improvement on a strong object detection baseline.

The overall architecture diagram includes the input image, the CNN backbone for extracting the visual feature pyramid, the introduced explicit visual center (EVC), the global centralized conditioning (GCR), and the decoupled head network for target detection, which consists of a classification loss, a regression loss, and a segmentation loss, where C denotes the class size of the dataset used. The CFP module structure is shown in Figure 4.

The specific flow of the CFP implementation is as follows. The input image is fed to the backbone network to extract a five-layer feature pyramid X, where the spatial size of each layer of feature $X_{i}$ is 1/2, 1/4, 1/8, 1/16, and 1/32 of the input image, respectively. The top layer of the feature pyramid (i.e., $X_{4}$) is captured using an EVC structure, and a proposedlightweight *MLP* architecture to capture the global long-range dependencies of $X_{4}$ (the lightweight *MLP* architecture is not only simpler in structure, but also lighter in size and more computationally efficient compared to a transformer encoder based on a multiple attention mechanism). A learnable visual center mechanism is used with the lightweight *MLP* to aggregate the input image’s local corner regions based on the proposed ECV, in order to enable the shallow features of the feature pyramid to simultaneously benefit from the visual centralization information of the deepest features in an efficient mode, where the explicit visual centralization information obtained from the deepest in-layer features is used to simultaneously modulate all the pre-shallow features (using GCR to modulate $X_{3}$ and $X_{2}$). These features are aggregated into a decoupled head network for classification and regression.

The EVC consists mainly of two blocks connected in parallel, lightweight *MLP* and LVC. The resultant feature maps of these two blocks are connected together along the channel dimension as the output of the EVC used for downstream recognition. Between $X_{4}$ and EVC, the stem block is used for feature smoothing instead of implementing it directly on the original feature maps. The stem block consists of a 7 × 7 convolution with an output channel size of 256, followed by a batch normalization layer and an activation function layer.

The lightweight *MLP* consists of two residual modules: a depth-separable convolution-based module and a channel *MLP*-based module. The input of the *MLP* module is the output of the depth-separable convolution module. Both modules undergo channel scaling and DropPath operations to improve feature generalization and robustness. Compared with spatial *MLP*, channel *MLP* not only effectively reduces the computational complexity, but also meets the requirements of general-purpose vision tasks. Finally, both modules implement channel scaling, DropPath and residual join operations. LVC is an encoder with an intrinsic dictionary consisting of an intrinsic codebook ($B=\{b_{1},b_{2},\ldots ,b_{k}\}$, where N=H×W is the total spatial number of input features, where H and W denote the spatial magnitude of the height and width of the feature maps, respectively) and a set of learnable visuocentric scale factors $\left(S=\{s_{1},s_{2},\ldots ,s_{k}\right)$. The processing of LVC consists of two main steps; first, encoding the input features using a set of convolutional layers and further processing using CBR blocks, and second, combining the encoded features with the intrinsic codebook through a set of learnable scale factors. For this purpose, we used a set of scale factors s in sequence such that $x_{i}$ and $b_{k}$ mapped the corresponding position information. The information about the *k*th codeword in the whole image can be computed in the following way.
$$e_{k}=\sum_{i=1}^{N}\frac{e^{{-s_{k}\left|\left|{\overset{ˇ}{x}}_{i}-b_{k}\right|\right|}^{2}}}{\sum_{j=1}^{K}e^{{-s_{j}\left|\left|{\overset{ˇ}{x}}_{i}-b_{j}\right|\right|}^{2}}}(\overset{ˇ}{x_{i}}-b_{k})$$
where $x_{i}$ is the *i*th pixel point, $b_{k}$ is the *k*th learnable visual codeword, $s_{k}$ is the *k*th scale factor also set as a learnable parameter. $\left({\overset{ˇ}{x}}_{i}-b_{k}\right)$ is information about the position of each pixel relative to the codeword. *K* is the total number of visual centers. A fully connected layer and a (1×1) convolutional layer are then used to predict salient key class features. Finally, the input features from the Stem block $X_{in}$ and the localized corner region features with scale factor coefficients are subjected to channel multiplication and channel addition.

The contribution of CFP is to propose an intra-layer feature conditioning method for the feature pyramid and a top-down global centralized conditioning strategy, which provides a more effective means of information extraction and feature optimization for the target detection task. The introduction of this algorithm injects innovation into the research framework of the thesis and provides strong support for improving target detection performance.

### 2.3. MultiFuseYOLO: A Study of Multisource Information Fusion Methods

In order to further improve the accuracy and robustness of grape variety identification, we added a novel combinatorial discriminative method of multisource information fusion to the WineYOLO-RAFusion model, and obtained a new model, MultiFuseYOLO, which aims to achieve more reliable varietal identification by combining optimized leaf and fruit information with effective processing of multitarget detection results. MultiFuseYOLO inherits the state-of-the-art architecture of the WineYOLO-RAFusion model, but incorporates a multisource information fusion mechanism in the final stage of the model. The core algorithm of the multisource information fusion mechanism is the SynthDiscrim algorithm, which uses a user-adjustable threshold that triggers the consideration of composite discriminations when the probability of classifying a fruit or leaf falls below the threshold. The flowchart of the multisource fusion method is shown in Figure 5.

#### 2.3.1. SynthDiscrim Algorithm

In the integrated discrimination process, multiple leaves and fruits detected in the image are first integrated. The integrated discrimination scores of leaves and fruits are obtained by accumulating the classification probabilities of all leaves and fruits for each variety $S_{z}$. It is assumed that for a particular variety, there are N leaf and M fruit detections corresponding to the classification probabilities of $s_{l1},s_{l2},...,s_{lN}$ and $s_{f1},s_{f2},...,s_{fM}$, respectively. Then, the average of the classification scores for fruits and leaves, denoted as $\bar{s_{f}}$ and $\bar{s_{l}}$, respectively, is calculated to combine the effects of the two, and this discrimination process is shown as follows.
$$S_{z}=\sum_{i=1}^{N}s_{li}+\sum_{j=1}^{M}s_{fj}$$

$\bar{s_{f}}$ The formulas for $\bar{s_{l}}$ are shown below:$$\bar{s_{l}}=\frac{1}{N}\sum_{i=1}^{N}s_{li}$$
$$\bar{s_{f}}=\frac{1}{M}\sum_{j=1}^{N}s_{fj}$$

After obtaining the composite judgment score $S_{z}$ for leaves and fruits, a final composite score S is obtained by multiplying the average classification scores of fruits and leaves by the corresponding weights. S is calculated using the following formula,
$$S=w_{f}\times \bar{s_{f}}+w_{l}\times \bar{s_{l}}$$
where S denotes the final composite judgment score of grapes in the image. $w_{f}$ and $w_{l}$ denote the weights used for fruit and leaf classification scores, respectively. $\bar{s_{f}}$ and $\bar{s_{l}}$ denote the average fruit and leaf classification scores, respectively.

#### 2.3.2. EMASlideLoss Function

EMASlideLoss uses exponential moving average (EMA) to smooth SlideLoss. EMA is an averaging method that gives more weight to the most recent data and progressively less weight to older data. In the formula, α is a parameter called the smoothing coefficient, which determines the weight of the newer observations relative to the older ones.
$$EMA_{t}=\alpha \cdot Loss_{t}+\left(1-\alpha \right)\cdot EMA_{t-1}$$
where $EMA_{t}$ is the EMA at time *t*, $Loss_{t}$ is the SlideLoss at time t, and α is the smoothing factor, usually between 0 and 1, which determines the weight of the new data.

The role of EMA allows for better smoothing, and by introducing EMA, EMASlideLoss aims to reduce fluctuations in the loss values. This helps the model to learn more consistently, especially when noise or outliers occur during training, and it also provides good generalization ability. Smoothing the loss values may help the model to generalize better to unseen data, as it relies more on stable trends rather than local noise. At each training step t, EMASlideLoss calculates the current SlideLoss value $Loss_{t}$, and then updates the EMA value using the EMA formula. The EMA value is considered as the smoothed loss and is used to guide the update of the model parameters. This approach prevents the model from being overly sensitive to outliers in the training set and improves the stability of the model. As the parameter α moves closer to 1, the new observations have a greater effect on the EMA and the EMA is more sensitive. α The closer to 0, the smoother the EMA is and the more resistant it is to noise and fluctuations.

Overall, the goal of EMASlideLoss is to improve the training of deep learning models by introducing exponential moving averages to improve the stability and generalization of training.

## 3. Experimental Results and Discussion

### 3.1. Experimental Platform

This experiment used Ubuntu 20.04.4 LTS 64 as the operating system (Canonical Ltd., London, UK) and Intel€ X€ (R) Silver 4214 as the processor with a CPU@2.20 GHz, and 32 G of RAM (Intel, Santa Clara, CA, USA). The GPU was an NVIDIA Tesla T4 with 16 G of video memory (Nvidia, Santa Clara, CA, USA). The program-mining language was Python and the PyTorch deep learning framework was used.

### 3.2. Evaluation Indicators

In this study, we used precision, recall, accuracy, and the F1 score as evaluation metrics. We used the values of these evaluation metrics to evaluate the model in a comprehensive manner. Precision, recall, accuracy, and the F1 score were calculated as follows.
$$Precision=\frac{TP}{TP+FP}$$
$$Recall=\frac{TP}{TP+FN}$$
$$F1=\frac{2TP}{2TP+FP+FN}$$
where TP is the number of true positive samples, TN is the number of true negative samples, FP is the number of false positive samples, and FN is the number of false negative samples. Accuracy is the ratio of the number of correctly predicted samples to the total number of samples used for model experiments. Precision is the ratio of the number of correctly predicted positive samples to the total number of correctly predicted samples, and recall is the ratio of the number of correctly predicted positive samples to the total number of positive samples. The F1 score is a comprehensive evaluation index that considers precision and recall, and the average of precision and recall.

### 3.3. Ablation Experiment

#### 3.3.1. Modifications to the Model

We conducted a series of ablation experiments in order to deeply evaluate the impact of introducing the Res-Attention module and the CFP module on the performance of target detection in YOLOv7. YOLOv7 is a target detection model with good performance and speed. We aimed to further improve its performance by adding the Res-Attention module and the CFP module. In our experiments, we performed 200 epochs of training to ensure that the model was fully learned on the public fruit dataset. In terms of data preprocessing, we adjusted and processed the images appropriately to fit the input requirements of the model. After the model training was completed, we evaluated the performance on the test set.

First, we designed an ablation experiment by adding the Res-Attention module only to YOLOv7. The Res-Attention module reduces information loss during information transmission by introducing residual structure and better preserves and fuses key features through channel-attention and spatial-attention models. We used the trained YOLOv7 model and embedded the Res-Attention module into the network structure, which was then evaluated on the fruitful dataset.

Second, we conducted another ablation experiment by adding the CFP module only to YOLOv7. The CFP module is a feature fusion module designed to improve the model’s ability to perceive features at different scales, thereby improving the accuracy and stability of the detection results. We also used the trained YOLOv7 model and integrated the CFP module into the network, and then evaluated it on the same dataset.

Finally, we conducted a set of integrated experiments where both the Res-Attention module and the CFP module were added to YOLOv7. The experimental results are shown in Table 4.

Through these ablation experiments, we were able to progressively evaluate the impact of the Res-Attention module and the CFP module on the performance of the YOLOv7 model and compare the effects when they were added individually and in combination to better understand the role of these modules in the target detection task.

From the above experimental results, we observed significant performance improvement when only the Res-Attention module was added to the YOLOv7 model. mAP@0.5 improved from the original 0.673 to 0.734, which indicates that the Res-Attention module played a positive role in improving the detection performance. By introducing the residual structure as well as the channel attention and spatial attention models, the Res-Attention module reduced the information loss during the information transmission process and better preserved and fused the key features, thus improving the detection accuracy. When only the CFP module was added to the YOLOv7 model, we similarly observed a significant increase in performance. mAP@0.5 increased from the original 0.673 to 0.748, which indicates that the introduction of the CFP module effectively improved the model’s ability to perceive features at different scales, which in turn improved the accuracy and stability of the detection results.

Finally, when both the Res-Attention module and the CFP module were added to the YOLOv7 model, we saw further performance improvements. Considering the effects of the two modules together, the model’s mAP@0.5 reached 0.763, with precision of 0.735, recall of 0.708, and F1 of 0.721. This indicates that the two modules synergized with each other when used in an integrated manner, which further enhanced the model’s detection performance.

Through the ablation experiments we clearly saw the important roles of the Res-Attention module and the CFP module in improving the performance of the YOLOv7 model. They improved the detection accuracy and stability of the model by reducing the information loss and improving the multiscale feature perception ability, respectively. When the two modules were applied simultaneously, they complemented each other and produced synergistic effects, further improving the model performance.

#### 3.3.2. Modifications to the Loss Function

To further evaluate the performance of the target detection model YOLOv7, we designed a series of ablation experiments by replacing the default loss function of YOLOv7 with EMASlideLoss and analyzing it in comparison with the original YOLOv7 default loss function. Next, we describe the experimental design. We first trained the model based on the original YOLOv7 model using EMASlideLoss as the loss function during training. Then, on the same dataset, we kept all other conditions constant and trained using the default loss function of YOLOv7, respectively. After the training was completed, we evaluated the performance of both models using the same evaluation metrics, including mAP@0.5, precision, recall, and F1 values. The same fruitful public dataset was used as in the above experiments and training for 200 epochs was performed. The experimental results are shown in Table 5.

The experimental results showed that the performance of the model using EMASlideLoss as the loss function was significantly better than that of the model using only the YOLOv7 default loss function. Specifically, the model using EMASlideLoss achieved an increase of 0.032 in the mAP@0.5 metric from 0.763 to 0.795, which indicates that EMASlideLoss played a positive role in improving the overall accuracy of the model for the target detection task. As for precision, recall, and F1 values, although precision remained unchanged, recall and F1 values were significantly improved, by 0.1 and 0.048, respectively, which suggests that EMASlideLoss not only improved the model’s accuracy, but also enhanced its detection of targets, especially in terms of identifying more comprehensively the difficult-to-detect targets or target groups. In the target detection task, the adoption of EMASlideLoss as a loss function effectively enhanced the performance of the model, resulting in significant improvements in different metrics.

### 3.4. Independent Grapevine Variety Identification Experiments

#### 3.4.1. Experimental Design

The aim of this experiment was to perform independent grapevine variety detection using the WineYOLO-RAFusion model, YOLOV5 [41], YOLOX [42], and YOLOV7. We chose the Embrapa WGISD public dataset, which contains images of several grape varieties and provides sufficient diversity for evaluating the varietal differentiation performance of the models.

For the experiments, we performed 200 epochs of training to ensure that the model fully learned on the dataset. For data preprocessing, we adjusted and normalized the images appropriately to fit the input requirements of the model. After the model training was completed, we carried out performance evaluation on the test set, focusing on the detection accuracy and generalization ability of the model on different grape varieties.

The key steps of the experiment included data preparation, model training, model evaluation, and result analysis. We carefully recorded the detection results of each grape variety and comprehensively evaluated the model performance with the help of precision, recall, F1 score, and other indicators. Finally, we analyzed the experimental results to explore the performance of the WineYOLO-RAFusion model on different grape varieties, with a special focus on its strengths and weaknesses in varietal differentiation. This experimental design aimed to provide detailed information for the performance evaluation of grapevine variety identification models, which will provide strong guidance for future improvement and application.

#### 3.4.2. Experimental Results and Analysis

The experimental results clearly demonstrated the excellent performance of the WineYOLO-RAFusion model in the fruit localization task, which significantly outperformed other models such as YOLOV5, YOLOX, and YOLOV7. The performance advantage of the WineYOLO-RAFusion model is attributed to its well-designed Res-Attention mechanism as well as the CFP centrality feature, and the introduction of the pyramid multiscale fusion module. The experimental results are shown in Figure 6.

The WineYOLO-RAFusion model achieved the best experimental results on multi-species fruit detection, showing its significant advantages in adapting to complex scenarios and diverse fruit varieties. In contrast, models such as YOLOV5, YOLOX, and YOLOV7 failed to reach the high level of WineYOLO-RAFusion, which may be due to the innovative design of WineYOLO-RAFusion in terms of Res-Attention and CFP modules, which helped to improve the accurate detection of fruits of different scales and complexity.

Overall, the comprehensive performance of the WineYOLO-RAFusion model excelled in the fruit localization task, which further demonstrates the importance of careful design of deep learning models through the introduction of advanced attention mechanisms and multiscale fusion modules. The results of this experiment provide a strong guideline for the field of fruit variety recognition, emphasizing that careful and effective architectural design is crucial for improving model performance.

Based on the comprehensive evaluation metrics of the experimental data, we compared the performance of four models, YOLOV5, YOLOX, YOLOV7, and WineYOLO-RAFusion, in the fruit localization task. The experimental results are shown in Table 6. The results showed that the WineYOLO-RAFusion model achieved the best performance on all evaluation metrics. Specifically, WineYOLO-RAFusion achieved an average precision (mAP@0.5) of 0.795, significantly outperforming YOLOV5 (0.633), YOLOX (0.654), and YOLOV7 (0.673). In terms of accuracy, WineYOLO-RAFusion performed the best, with a precision (Precision) of 0.735, which was much higher than that of the other models. Meanwhile, WineYOLO-RAFusion also excelled in comprehensiveness, with a recall of 0.808. Taken together, WineYOLO-RAFusion achieved an F1 score of 0.769, once again highlighting its all-around superiority in the fruit localization task. This set of data results further confirmed the successful design of the WineYOLO-RAFusion model introducing the Res-Attention and CFP-centered feature pyramid multiscale fusion modules, which give it excellent performance and robustness in multi-species fruit detection.

The experimental results showed that Cabernet Franc, Cabernet Sauvignon, and Syrah were relatively well recognized in the category recognition task, reaching high accuracies of 0.814, 0.923 and 0.893, respectively. However, when facing the two varieties of Chardonnay and Sauvignon Blanc, the accuracy of the category recognition was relatively low, 0.512 and 0.533, respectively, because their fruit shapes were too similar for the traditional fruit characteristics to make effective distinction. The results of comparing the precision of each variety on the WineYOLO-RAFusion independent fruit dataset are specifically shown in Table 7.

To address this challenge, we decided to introduce leaf characteristics as an adjunct to improve the identification accuracy of Chardonnay and Sauvignon Blanc. This strategy aimed to synthesize information from both fruits and leaves, taking full account of the multiple sources of plant characteristics. This integrated approach was expected to improve the reliability of category identification while addressing the situation of high varietal similarity. To address this challenge, we decided to introduce leaf characteristics as an adjunct to improve the identification accuracy of Chardonnay and Sauvignon Blanc. This strategy aimed to synthesize information from both fruits and leaves, taking full account of the multiple sources of plant characteristics. This integrated approach was expected to improve the reliability of category identification while addressing the situation of high varietal similarity.

### 3.5. Independent Grape Leaf Variety Identification Experiments

#### 3.5.1. Experimental Design

The main goal of the experiment was to improve the classification accuracy of these similar varieties by performing varietal identification on the leaves, with a particular focus on varieties with similar fruit appearance, such as Chardonnay and Sauvignon Blanc. During the data collection phase, we used a selfie stick and a Sony lens to photograph the leaves of grape plants at a distance of 20–30 cm to ensure that the leaf images contained sufficient detail and diversity to provide adequate training samples for the model.

For model training and evaluation, we continued to use the WineYOLO-RAFusion model with similar training parameters and evaluation metrics. During the training process, the focus was on learning the leaf-specific texture and morphology. Once the model training was complete, it was evaluated with a test set focusing on the identification accuracy of individual grape varieties. The number of training rounds was 200 epochs.

With this experimental design, we expected to improve the classification performance of grapevine varieties through the comprehensive utilization of leaf characteristics, especially in the face of high fruit similarity. This will provide more valuable information for a comprehensive understanding of grapevine plants and a more in-depth perspective for research and applications in variety identification.

#### 3.5.2. Experimental Results and Analysis

From Table 8, which contains experimental results, it can be observed that leaf recognition alone did not perform as well as the fruit recognition alone experiment. This may be attributed to the fact that the features of leaves are relatively more complex and more difficult for the model to accurately recognize.

Fruits usually have more obvious and unique features such as shape and color, which are easy for models to learn and identify. In contrast, the shape and texture of leaves may be more detailed and susceptible to factors such as light and angle, making it more difficult for models to recognize them.

Observing the results of the leaf recognition experiment, it was found that it performed better in the recognition of two varieties, Chardonnay and Sauvignon Blanc, with a precision of 0.81 for Chardonnay and 0.83 for Sauvignon Blanc, which was a relatively high precision. However, the recognition of the other four varieties was poor, resulting in an overall accuracy that was not very high. This was in contrast to the performance in the previous fruit recognition experiments.

Chardonnay and Sauvignon Blanc had the worst recognition accuracy in the fruit recognition experiment, while they performed relatively well in the leaf recognition experiment. This may indicate that leaf characteristics are more favorable for the identification of these two varieties, whereas fruit appearance characteristics are relatively more difficult to distinguish.

By analyzing the results of separate experiments for both fruit and leaf, we observed some interesting trends. Specific varieties (e.g., Chardonnay and Sauvignon Blanc) were relatively less accurately recognized in the fruit recognition experiments, while both varieties were relatively better recognized in the leaf recognition experiments.

This observation emphasizes the possible differences in grapevine plants for variety identification at different growth stages or when different characteristics are used. Since fruit and leaf characteristics each have their own strengths, it may be more beneficial to combine the characteristics of both to achieve a more comprehensive and accurate variety identification. In practical applications, we may consider combining information from both fruit and leaves to fully utilize the multiple sources of plant characteristics.

This integrated approach is expected to improve the robustness of varietal identification, especially when confronted with varieties that are similar in appearance. By combining fruit and leaf characteristics, we can expect more comprehensive and accurate grape variety identification results, providing a more reliable basis for decision-making and further research.

### 3.6. MultiFuseYOLO Co-Detection Experiment: Common Identification of Grapevine Fruit and Leaf Varieties

#### 3.6.1. Experimental Design

The purpose of this experiment was to evaluate in-depth the performance of the MultiFuseYOLO method in a grape variety identification task against target detection models such as YOLOV5, YOLOX, YOLOV7, and WineYOLO-RAFusion. We used a publicly available grape variety identification dataset as a test subject to ensure diversity in the experimental benchmarks. During the training process, in order to better adapt to the multisource information fusion characteristics of MultiFuseYOLO, we used the public dataset for fruit training, and introduced the self-harvested leaf dataset for leaf training. Attention was paid to MultiFuseYOLO’s integrated processing of fruit and leaf information, as well as the accurate discrimination of various grape varieties. The other comparison models, YOLOV5, YOLOX, YOLOV7, and WineYOLO-RAFusion, continued to be compared using the traditional methodology, which still employed the results generated on the publicly available fruit dataset as described above.

After several experiments, we adjusted and determined the fruit weight $W_{f}$ and leaf weight $W_{l}$ in the SynthDiscrim algorithm. When dealing with varieties with high fruit similarity and high leaf variability, we found that setting $W_{f}$ to 0.35 and $W_{l}$ to 0.65 worked best. Such a setting enabled MultiFuseYOLO to obtain the best results when combining fruit and leaf information, thus achieving the highest recognition precision. Therefore, for all subsequent experiments on MultiFuseYOLO, we set the fruit weights $W_{f}$ and leaf weights $W_{l}$ to the above results.

#### 3.6.2. Experimental Results and Analysis

Table 9 demonstrates in detail the excellent performance of the MultiFuseYOLO method in the grape variety identification task. A precision of 0.854, a recall of 0.815, and an F1 score of 0.833 demonstrated the excellent performance of the method in wine grape detection. This was especially true when addressing Chardonnay and Sauvignon Blanc, two varieties that are difficult to distinguish from each other.

From Table 10 we can see the method’s precision of 0.813 and 0.775 for Chardonnay and Sauvignon Blanc, respectively, underscoring the excellence of MultiFuseYOLO in solving the problem of recognizing these cosmetically similar varieties. By utilizing a combination of fruit and leaf information, MultiFuseYOLO not only achieved a significant advantage in ensuring the accuracy of the model’s prediction results, but also demonstrated its excellent performance in terms of recall and overall model performance with its high performance in Recall and F1 scores.

The MultiFuseYOLO method not only achieved a significant improvement in Precision, emphasizing its focus on ensuring the accuracy of the model’s prediction results, but also demonstrated its superiority in terms of recall and overall model performance with its high performance in recall and F1 scores. This is critical for tasks such as grape variety identification that require a high degree of accuracy and overall performance.

These significant advantages displayed in the experimental results endow MultiFuseYOLO with a leading position in the field of grape variety identification. The method not only provides reliable support for agricultural decision-making and management, but also provides a strong empirical basis for future related research. Its successful application on the fusion of multisource information makes MultiFuseYOLO the model of choice for practical applications, providing a more powerful and sustainable solution for intelligent decision-making in agriculture.

## 4. Conclusions

In this study, we explored the task of identifying wine grape varieties. Through separate experimental analyses of fruit and leaf characteristics, we found that certain wine grape varieties posed greater difficulties when identified and classified based on fruit or leaf characteristics alone. Recognizing this challenge, this paper aimed to address this issue by improving the performance of grape variety identification models at the model level.

To address this problem, we developed a highly specialized target detection model, WineYOLO-RAFusion, based on the YOLOV7 model, which has been carefully designed to excel in classifying wine grape varieties. However, since WineYOLO-RAFusion has difficulty in distinguishing between wine grape varieties that are highly similar, a new model, MultiFuseYOLO, was proposed based on this problem. MultiFuseYOLO is a new model obtained by adding a multisource information fusion method to the WineYOLO-RAFusion model. The core of the multisource information fusion method is the SynthDiscrim algorithm, which seamlessly integrates fruit and leaf information using complex techniques such as weighted feature fusion, optimized threshold setting and combined discriminant score calculation. These improvements greatly enhanced the model’s ability to accurately recognize various grape varieties.

On the experimental side, we rigorously compared the model proposed in this paper with other state-of-the-art models (e.g., YOLOV5, YOLOX, YOLOV7, and WineYOLO-RAFusion), and the results highlighted the superior performance of MultiFuseYOLO in terms of precision, recall, and F1 score. Notably, the model exhibited extremely high accuracy, especially in recognizing similar-looking species and achieving high-precision recognition.

In conclusion, the focus of this study was to improve the performance of grape variety identification at the model level. The innovation of MultiFuseYOLO by combining the WineYOLO-RAFusion model with the multisource information fusion approach is a major advancement in the accurate identification of wine grape varieties. These contributions provide powerful and effective solutions to the challenges encountered in wine grape variety identification.

## Figures and Tables

**Figure 1 sensors-24-02953-f001:**
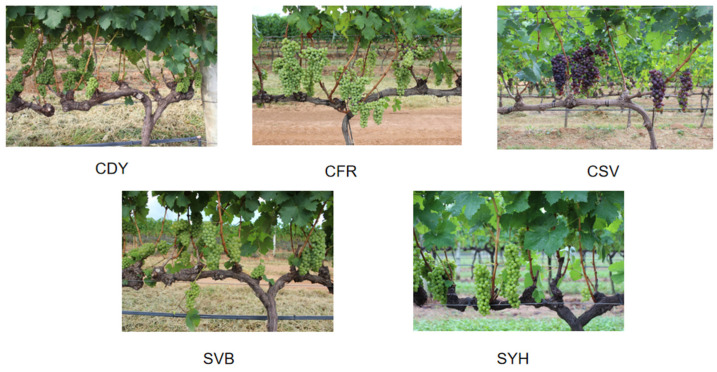
Embrapa WGISD species map.

**Figure 2 sensors-24-02953-f002:**
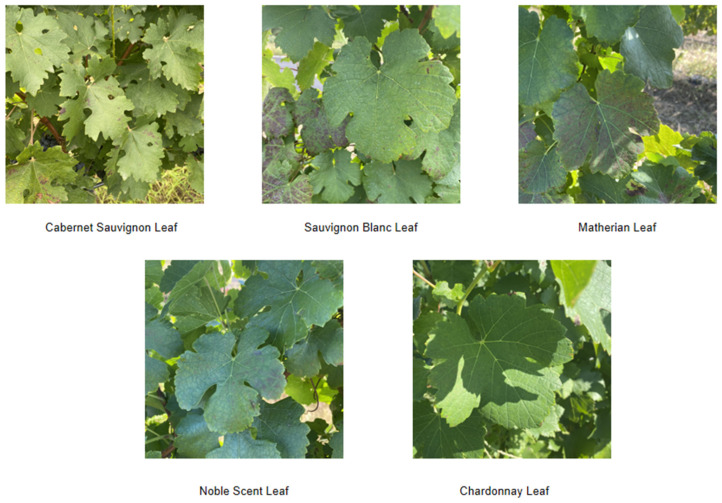
Self-mined blade dataset.

**Figure 3 sensors-24-02953-f003:**
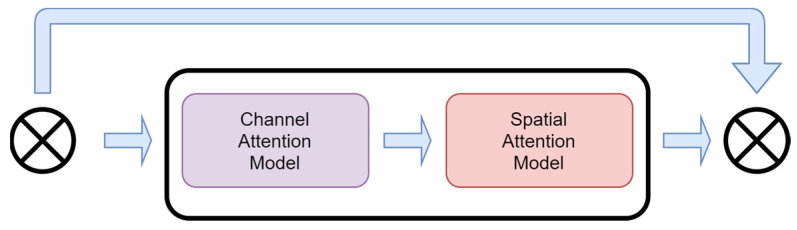
Structure of Res-Attention module.

**Figure 4 sensors-24-02953-f004:**
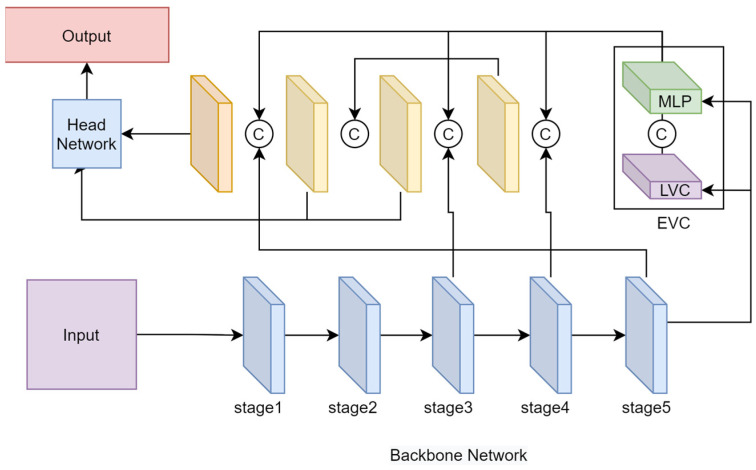
CFP structure diagram.

**Figure 5 sensors-24-02953-f005:**
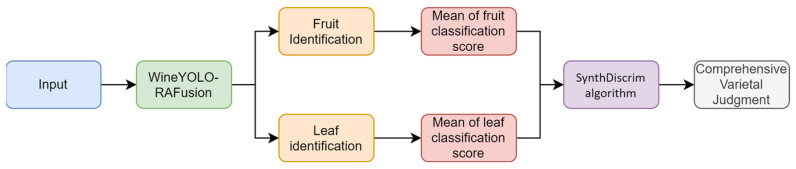
Flowchart of multisource fusion method.

**Figure 6 sensors-24-02953-f006:**
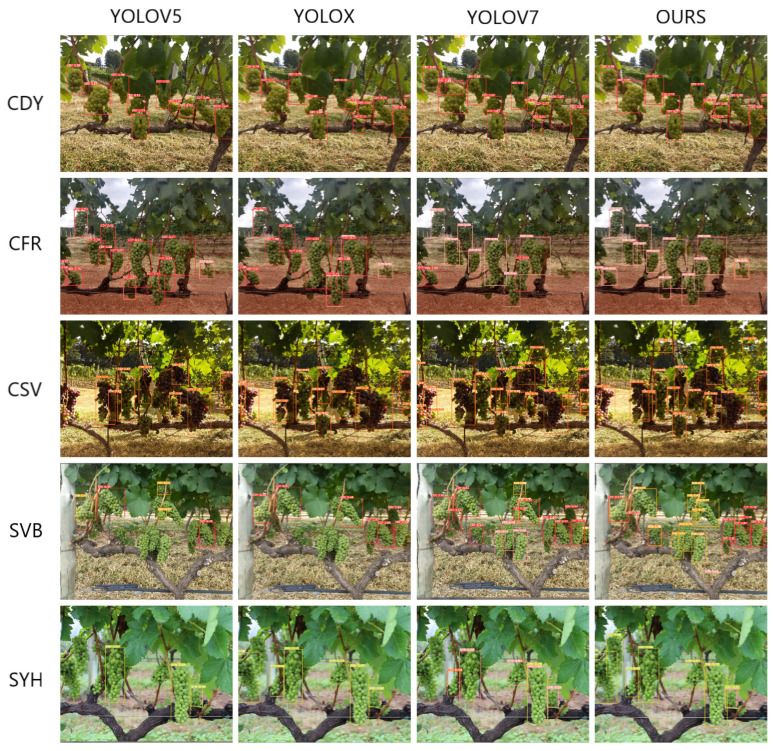
Plot of detection and localization results for each model.

**Table 1 sensors-24-02953-t001:** Embrapa WGISD [23].

Acronyms	Variety Name	Image
CDY	Chardonnay	65
CFR	Cabernet Franc	65
CSV	Cabernet Sauvignon	57
SVB	Sauvignon Blanc	65
SYH	Syrah	48
Total		300

**Table 2 sensors-24-02953-t002:** Camera detailed parameters.

Parameter Name	Parameter Details
Focus mode	Auto/Manual
Sensor chip	Coms 1/2.8 inch
Output resolution	4656 × 3496
Connector/driver	USB/Driverless
Frame rate	10/frame (supports 30 fps)
Cable length	2 m/1.5 m
Exposure	Automatic Adjustment
Voltage/current	5 V/150
Output format	Mjpg/yuy2
Angle size	65–85 degrees

**Table 3 sensors-24-02953-t003:** Self-harvested leaf dataset.

Assortment	Dates	Imagery
Cabernet Sauvignon Leaf	2023.10	500
Sauvignon Blanc Leaf	2023.10	500
Matherian leaf blades	2023.10	300
Noble Scent Leaf	2023.10	300
Chardonnay Leaf	2023.10	300

**Table 4 sensors-24-02953-t004:** Model modification experimental results.

Mold	mAP@0.5	Precision	Recall	F1
YOLOV7	0.673	0.689	0.654	0.669
YOLOV7 + Res-Attention	0.734	0.712	0.677	0.695
YOLOV7 + CFP	0.748	0.724	0.701	0.712
WineYOLO-RAFusion (YOLOV7 + Res-Attention + CFP)	0.763	0.735	0.708	0.721

**Table 5 sensors-24-02953-t005:** Loss function comparison experimental results.

Mold	mAP@0.5	Precision	Recall	F1
WineYOLO-RAFusion	0.763	0.735	0.708	0.721
WineYOLO-RAFusion + EMASlideLoss	0.795	0.735	0.808	0.769

**Table 6 sensors-24-02953-t006:** Comparison of WineYOLO-RAFusion with each model on separate fruit datasets.

Mold	mAP@0.5	Precision	Recall	F1
YOLOV5	0.633	0.642	0.522	0.567
YOLOX	0.654	0.672	0.602	0.636
YOLOV7	0.673	0.689	0.654	0.669
WineYOLO-RAFusion	0.795	0.735	0.808	0.769

**Table 7 sensors-24-02953-t007:** Comparison of precision of individual varieties on WineYOLO-RAFusion separate fruit dataset.

Kind	Precision
Chardonnay.	0.512
Cabernet Franc	0.814
Cabernet Sauvignon	0.923
Sauvignon Blanc	0.533
Syrah	0.893

**Table 8 sensors-24-02953-t008:** Comparison of WineYOLO-RAFusion with each model on separate leaf datasets.

Mold	Precision	Recall	F1
YOLOV5	0.588	0.517	0.550
YOLOX	0.601	0.612	0.606
YOLOV7	0.615	0.599	0.607
WineYOLO-RAFusion	0.723	0.693	0.708

**Table 9 sensors-24-02953-t009:** Comparison of MultiFuseYOLO with the other models.

Mold	Precision	Recall	F1
YOLOV5	0.642	0.522	0.567
YOLOX	0.672	0.602	0.636
YOLOV7	0.689	0.654	0.669
WineYOLO-RAFusion	0.735	0.808	0.769
MultiFuseYOLO	0.854	0.815	0.833

**Table 10 sensors-24-02953-t010:** Comparison of MultiFuseYOLO’s single precision for each species in the public dataset.

Kind	Precision
Chardonnay	0.813
Cabernet Franc	0.852
Cabernet Sauvignon	0.931
Sauvignon Blanc	0.775
Syrah	0.899

## Data Availability

Data are available on request due to privacy.

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
