# Peer review of "MultiFuseYOLO: Redefining Wine Grape Variety Recognition through Multisource Information Fusion"

_sensors, 2024, doi:10.3390/s24092953_

Round 1

Reviewer 1 Report

Comments and Suggestions for Authors

1. Abstract line 18.   It is recommended that the authors clarify which traditional methods.

2. Line 23-24. Although the recognition accuracy of Chardonnay was improved from 0.501 to 0.533, the recognition accuracy of Sauvignon Blanc varieties was reduced from 0.813 to 0.775, and the accuracy was not improved. In addition, it is mentioned in the conclusion that the proposed method achieves high recognition accuracy, while the recognition accuracy of Chardonnay is only 0.533, could the authors please explain this in detail?

3. Abstract line 24.  Authors should pay attention to the harmonization of data.

4. Introduction. The introduction should focus on the significance and importance of the work of this paper thus providing the reader with ideas for research rather than going into great detail about the work of this paper. The third paragraph (lines 46-72) and the fourth paragraph (lines 73-82) do not seem to be logically related. It is recommended that the authors carefully consider this section.

5. Lines 101-102. Please pay attention to the writing conventions of the paper.

6. Please pay attention to the citation of figures and the uncited figures should be removed from the manuscript.

7. Please describe in detail the mechanism of the Res-Attention module and the CFP-centered feature pyramid  multiscale fusion module  in YOLOv7 for this study.

8. As far as I know, the version of the YOLO has been updated to v9, do the authors evaluated wine varietal identification using the latest version?

9. Figure 6. It is clear that the proposed method has leakage in variety SYH, which is fully recognized by the baseline model yolov7. Please explain this phenomenon.

Reviewer 2 Report

Comments and Suggestions for Authors

The article is devoted to a rather interesting problem of segmentation and classification of grape images. The authors use the modified YOLOv7 model, expanding its capabilities through training and the use of additional modules, as well as a combined approach when the analysis is not only for fruits, but also for leaves. Each module of the author's approach is described in sufficient detail. The experimental part is performed at a high level. First, the authors prove the effect of each added module in the ablation experiment. Next, they evaluate the quality of image segmentation in comparison with the original and alternative models. Finally, a comparison of the author's MultiFuseYOLO approach with existing methods is presented. All of the above allows you to give a positive conclusion on the submitted work

However, there are opportunities for improvement and refinement in any work. I have noted the following points that are worth revealing in more detail:

1) Line 114 - the characteristics of the camera are not very clear, but the most important thing is the resolution of the images from which the dataset is assembled.

2) Figure 4 requires an extended description of all the abbreviations used, the layers stage1-stage5 are also not fully understood, it is not entirely clear from the text what these layers or modules are. 

3) Line 273, however, the remark is more relevant to section 3.5. The authors introduce a formula with two weight coefficients, but it is not disclosed in the experimental section

Round 2

Reviewer 1 Report

Comments and Suggestions for Authors

The authors have provided an adequate response to the comments indicated in the first review. Anyway, it recommends to use the high-resolution figures in the manuascript.

Author Response

Dear reviewer,

thank you for your affirmation, I have replaced all the figures with higher resolution ones. Finally thank you again.